# Sustainable Intensification with Cereal-Legume Intercropping in
Eastern and Southern Africa

**DOI:** 10.3390/su11102891

**Published:** 2019-05-21

**Authors:** Abednego Kiwia, David Kimani, Rebbie Harawa, Bashir Jama, Gudeta W. Sileshi

**Affiliations:** 1Alliance for Green Revolution in Africa (AGRA), West End Towers, Waiyaki Way, P.O. Box 66773 Westlands, Nairobi 00800, Kenya; AKiwia@agra.org (A.K.); DKimani@agra.org (D.K.); RHarawa@agra.org (R.H.); 2Islamic Development Bank, 8111 King Khalid St., Al Nuzlah Al Yamania Dist., Jeddah 22332-2444, Saudi Arabia; BAdan@isdb.org; 3Plot 1244, Ibex Hill, 10100 Lusaka, Zambia; 4School of Agricultural, Earth and Environmental Sciences, University of KwaZulu-Natal, Pietermaritzburg 4041, South Africa

**Keywords:** agronomic efficiency, climate-smart agriculture, mother-baby trial, nutrient mining, trade-off

## Abstract

Much research has been conducted on cereal-legume intercropping as a sustainable
intensification (SI) practice in Eastern and Southern Africa (ESA). However, the
role of inorganic fertilizers in sustainably intensifying intercropping systems
has not been systematically analyzed. Therefore, the objective of the present
analysis was to assess the role of inorganic fertilizer use in cereal-pigeonpea
(*Cajanus cajan*) intercropping in terms of SI indicators,
namely, yield, production risks, input use efficiency, and economic returns. The
data used for this analysis were gathered from over 900 on-farm trials across
Kenya, Tanzania, and Mozambique. All SI indicators assessed showed that
intercropping combined with application of small amounts of inorganic
fertilizers is superior to unfertilized intercrops. Fertilizer application in
the intercropping system improved cereal yields by 71–282% and pigeon pea
yields by 32–449%, increased benefit–cost ratios by 10–40%,
and reduced variability in cereal yields by 40–56% and pigeonpea yields
by 5–52% compared with unfertilized intercrops. Improved yields and
reduced variability imply lowering farmers’ risk exposure and improved
credit rating, which could enhance access to farm inputs. We conclude that the
strategic application of small amounts of inorganic fertilizers is essential for
the productivity and economic sustainability of cereal-pigeonpea intercropping
under smallholder farming in ESA.

## Introduction

1.

The smallholder agricultural sector in Eastern and Southern Africa (ESA) region is
heavily constrainedbydecliningpercapitalandholding[1],lossofsoilfertility[2,3],andclimatevariability[4-6].
Asinmostofsub-SaharanAfrica, thisregionisseverelyaffectedbylanddegradation[7]. Severaldecades of farming without use
of fertilizer inputs, from both organic and inorganic sources, has stripped the
soils of the vital nutrients needed to support plant growth [2], and this is now posing unprecedented social, economic,
and environmental problems [7].
Sustainable intensification (SI) of smallholder agriculture has been recognized as a
crucial component of the strategy towards reversing the trend in land degradation
and increasing food production [1,8,9]. Sustainable intensification (SI) is defined as producing more output
from the same area of land while reducing the negative environmental impacts, and at
the same time increasing contributions to natural capital and the flow of
environmental services [9,10]. SI is now recognized as one of the
cornerstones of climate-smart agriculture (CSA), i.e., agriculture that achieves the
triple objectives of increasing productivity, adaptation to climate change, and
mitigating greenhouse gas emissions [11].

Cereal-legume intercropping has been recognized as one of the SI pathways because
intercropping provides greater stability than sole cropping (monocultures) in terms
of soil fertility improvement, yields enhancement, and financial returns [12,13]. Cereal-legume intercropping also ensures diversification of diets
and risk reduction in the case of failure of one of the crops [12]. Among the legumes, *Cajanus cajan* (L.)
Millsp (pigeonpea) has been shown to hold the greatest potential for intercropping
with cereals in ESA because of its drought-tolerance ability and the role it plays
in household nutrition, income, and enhancing productivity. Unlike most legumes,
pigeonpea is well adapted to semi-arid and arid regions, and is traditionally
intercropped with maize (*Zea mays* (L.)), sorghum (*Sorghum
bicolar* (L.) Moench), and millet (*Eleucine coracana*
(L.) Gaertn, ssp. *Coracana*) in ESA. For example, the Konso people,
living in an arid pastoral area of south-central Ethiopia, have for centuries
practiced pigeonpea intercropping with ratooned sorghum [14]. Similarly, in southern Malawi pigeonpea is
traditionally ratooned and managed as a perennial crop in maize cropping systems
[15]. The intercrop diversifies the
production system and helps to minimize risks associated with growing only one crop
[8]. Additionally, it can provide
opportunities for increased use efficiency of the production resources, including
water, and is thus a sound cropping system in drylands.

In areas prone to extreme weather conditions, intercropping of staple cereals with
pigeonpea has been reported to provide greater insurance against crop failure [8,16,17]. In addition, most
pigeonpea varieties mature during the dry season long after farmers have harvested
their cereal crops, thereby bridging the hunger period and providing food security
to rural households [18-20]. It also provides additional benefits,
such as fodder for livestock and firewood for household energy needs [19,21]. The firewood supply can have labor-saving benefits for women who
are traditionally expected to perform this task for the family. There could also be
environmental benefits resulting from reduced pressure for firewood on the
woodlands, even if only for short periods.

Due to its complementarities with cereals and its ability to fix nitrogen (N)
biologically from the atmosphere, the benefits of cereal-pigeonpea intercropping in
long-term soil fertility improvement have been well documented [16–18].
Across the semi-arid areas of ESA, pigeonpea is cultivated on over 400,000 ha [22], but the potential is much greater. The
rising and sustained import demand from India and the Middle East [22,23] is likely to increase the land area that could potentially be
brought under the crop. The challenge to improved trade prospects has, however, been
its low productivity, especially under smallholder cropping systems, which is the
dominant form of pigeonpea production [22]. Pigeonpea yields are typically under 1.0 t ha^−1^ in
these systems [22,23]. The yields could, however, be more than tripled with
the use of improved varieties and good agronomic practices, including the use of
small amounts of fertilizers or manure [24].

Much research has been done on cereal-pigeonpea intercropping on research stations,
comparing yields in intercrops with sole maize [18,25,26]. Most of the studies examined the primary trade-offs in
cereal-pigeonpea intercropping, which include competition [20,26,27] and nutrient mining [19]. For example, in a long-term study in
Zimbabwe [26], maize yields without
fertilizer were lower in the intercrop than the sole maize in five out of 12
seasons. Results from other studies conducted through a multi-location trials
approach in Tanzania and Malawi raise concerns about the long-term sustainability of
the production system based on the levels of nutrients removed with crop harvests
[19]. The annual net removals of N
and P were estimated at 31–49 kg ha^−1^ in Malawi and
35–68 kg ha^−1^ in Tanzania. For P, the estimates stood at
9–17 kg ha^−1^ and 6–25 kg ha^−1^,
respectively.

Several studies, including one that conducted meta-analysis of several on-farm trials
[12], show that cereal yields are
generally less than 3.0 t ha^−1^, and that a global consensus is
building around the minimum threshold required to achieve household food
self-sufficiency and kick-start a smallholder-led “green revolution”
in Africa [2]. This is particularly
important for maize, which is a staple food crop in many countries in sub-Saharan
Africa (SSA). Raising the yields is important given that when farmers move from
under 1.0 to 3.0 t ha^−1^ or more, they generally begin to diversify
their production to income-generating practices that also engender the
sustainability of the production system [2]. This level of productivity cannot be achieved under smallholder farm
conditions unless external inputs, such as fertilizer and manure, are applied.
However, the role of inorganic fertilizer in sustainably increasing productivity in
intercropping has not been systematically studied. Information is also virtually
lacking on the profitability of fertilizer use in cereal-pigeonpea intercropping
systems in much of the ESA.

Increasing fertilizer use in legume intercrops would require awareness creation,
given that the practice is not common among smallholder farmers in the ESA region
[27,28]. To this end, the Alliance for Green Revolution in
Africa (AGRA), through its Soil Health Program, supported several projects across
SSA. This involved establishing a large number of participatory trials on
farmers’ fields over 4–5 years, which allowed farmers to participate
actively in their design and management. In addition to demonstrating technologies,
the trials were used to generate yield and socio-economic data that could be used to
assess the performance of the cropping systems in various agro-ecologies.
Traditional analyses of such data focus on yield and yield components at the local
level. Given limitation in their scope, such analyses fail to provide the evidence
needed to guide policy and good practice. In order to draw broader conclusions
relevant to policy, the present work aimed at cross-country analysis to have an idea
of the sustainability of the practices.

Sustainability is a very broad concept encompassing natural, social, and economic
capital, and it would not be possible to present a single metric applicable to any
situation [9,10,29]. As such,
assessing the sustainability of the intercropping system in the conventional
sustainability framework will be a daunting task. Instead, in this analysis we
focused on comparing treatments using indicators of sustainable intensification,
which have been reported to be more relevant to the system, such as the one studied
below. Several indicators of SI are available in the literature, and these fall
under five major domains, namely productivity, environmental sustainability,
economic sustainability, social sustainability, and human wellbeing [29]. The main objective of this analysis
was to assess the role of fertilizer use in cereal-pigeonpea intercropping in terms
of agricultural productivity and economic indicators of SI, specifically yields,
production risks, input use efficiency, and economic returns on smallholder farms.
The results of this work are expected to lead to better understanding of the
interactions between soil infertility and the livelihoods of smallholder farmers.
This work is also aimed at providing practical solutions to address the key issue of
soil fertility depletion through nutrient mining, which constrains land productivity
and the well-being of farmers.

## Materials and Methods

2.

### Study Areas and Treatments

2.1

The data came from on-farm trials conducted by three projects supported by
Alliance for Green Revolution in Africa (AGRA) in Mozambique, Tanzania, and
Kenya. Details of the trial sites are presented in Table 1. In terms of their farming systems and
socio-economic context, the study areas are more or less homogenous. The farming
system is dominated by the maize mixed farming system. This farming system
covers 10% of the land area, including most of the plateau and highland areas of
East Africa (including Kenya and Tanzania) and southern Africa (including
Mozambique). It serves as the food basket, as well as the driver of agricultural
growth and food security in the region. However, this farming system is
currently in crisis due to several interacting factors, including decline in
average farm sizes, low use of agricultural inputs, loss of nutrients and
organic matter [2], greater poverty
levels, and vulnerability to climate change [4,5].

**Table 1 t0001:** The sites covered by the projects, their dominant soils, and amounts of N
and P fertilizer applied in the different countries.

Country	District/County	Agro-Ecology Dominant Soils ^[Table-fn tf1-1]^	Dominant Soils z	Fertilizer Dose ^[Table-fn tf1-2]^	N Rate (kg ha^−1^)	P Rate (kg ha^−1^)
Mozambique	Barue	Sub-humid	Ferralsols	Full	73, 145	30
				Half	15	30
	Gorongosa	Sub-humid	Lixisols	Full	73	30
				Half	15	30
	Moatize	Sub-humid	Lixisols	Full	80	20
				Half	17	10
	Tsangano	Sub-humid	Lixisols	Full	80	20
				Half	34	17
	Angonia	Sub-humid	Lixisols	Full	68	35
				Half	34	17
	Vanduzi	Sub-humid	Lixisol	Full	73	30
				Half	15	30
	Susundenga	Semi-arid	Lixisol	Full	73	30
				Half	15	30
	Manica	Semi-arid	Phaeozems	Full	73	30
				Half	15	30
Tanzania	Arumeru	Sub-humid	Leptosols	Full	60	20
	Hai	Sub-humid	Nitisols	Full	60	20
	Kondoa	Semi-arid	Leptosols	Full	60	20
	Moshi	Sub-humid	Nitisols	Full	60	20
	Siha	Sub-humid	Nitisols	Full	60	20
Kenya	Makueni	Semi-arid	Ferralsols	Full	60	20
				Half	30	10

‡The dominant soil types are strictly based on the harmonized soil
atlas of Africa following the World Reference Base for Soil
Resources (WRB) classification and correlation system. The United
States Department of Agriculture (USDA) class equivalents are:
Acrisols = *Ultisols*; Cambisols =
*Inceptisols*; Ferralsols =
*Oxisols*; Leptosols = *Entisols*;
Lixisols = *Alfisols*; Nitisols =
*Alfisols*.

†Full and Half represent the recommended and half or less than half of
the recommended N and P fertilizer dosages.

In Mozambique, the trials were conducted during 2012–2014 in Angonia,
Barue, Gorongosa, Moatize, and Tsangano districts. The sites are characterized
by a single growing season and a unimodal rainfall pattern, followed by a long
dry season that lasts for five months from December to April. In all the
countries, the treatments included cereal-pigeonpea intercropping without
fertilizer (used as the control in this analysis), sole cereal crop, and sole
pigeonpea combined with fertilizer. The NPK fertilizer 12:24:12 was applied as a
basal dressing and urea as top dressing. Sole pigeonpea receiving N (22.5 kg
ha^−1^) and P (20 kg ha^−1^) was included on
some sites. The N and P rates applied to the cereal varied with site (Table 1). On other sites, small N doses
(≤50% the recommended rate) were applied to maize (Table 1). For brevity, these are referred to as
“half” rate, while application rates exceeding this (≥50%
of recommended) are referred to as “full”. Hybrid maize was
intercropped with an improved medium-duration (150–200 days to maturity)
pigeonpea variety (ICEAP 0557) of an indeterminate growth habit and
water-limited yield potential of 3000 kg ha^−1^. The maize
varieties were PAN 53 and PAN 67, which have a water-limited yield potential of
9000–10,000 kg ha^−1^.

The sites in Tanzania are characterized by a bi-modal annual rainfall cycle with
the major rainy season (often called the long rains) occurring during
March–June and the “short rains” during
October–December. The trials were implemented during 2010–2012 in
seven districts: Hai, Moshi, and Siha in Kilimanjaro region; Kondoa Districts in
Dodoma region; and Arumeru District in Arusha region. The treatments included
maize-pigeonpea intercropping and sole maize combined with the full rate of
fertilizer, consisting of urea (60 kg N ha^−1^) and phosphorus
fertilizer in the form of either diammonium phosphate (DAP) (20 kg P
ha^−1^), Minjingu hyperphosphate (20 kg P
ha^−1^), or Minjingu Mazao (20 kg P ha^−1^).
Hybrid maize was intercropped with the same pigeonpea variety (ICEAP 0557) used
in Mozambique. The maize variety used was PAN 691 with a water-limited yield
potential of 8000 kg ha^−1^.

The trials in Kenya were conducted during 2012–2014 on farms spread across
Kwa Kakulu and Kavingoni locations in Makueni County in the eastern semi-arid
lowlands. The sites have bimodal rainfall, as in Tanzania. Makueni County is
largely semi-arid and usually prone to frequent droughts. The treatments
consisted of intercropping of sorghum (variety Gadam) with pigeonpea, sole
sorghum, and sole pigeonpea. The intercrops received either half or full rate of
the recommended fertilizer for sorghum. The N fertilizer was applied in the form
of calcium ammonium nitrate (CAN), while the phosphorus fertilizer was applied
in the form of DAP. The full rate of fertilizer consisted of 60 kg N
ha^−1^ and 20 kg P ha^−1^, while the half
rate was 30 kg N ha^−1^ and 10 kg P ha^−1^. An
improved pigeonpea variety (Mbaazi 2) with a water-limited yield potential of
3200 kg ha^−1^ was used on all sites. This variety has an
indeterminate growth habit, taking more than 220 days to mature. Planting date,
population densities, planting methods, weeding, and all other agronomic
practices were performed, as per extension recommendation for the specific study
areas.

### Study Design

2.2

SI relies on collaboration between researchers and farmers to develop locally
appropriate agricultural technologies [10]. In that spirit, the design involved the mother-baby
participatory trial concept, which systematically links a central “mother
trial” managed by researchers to numerous farmer-managed
“baby” trials [28]. The
central mother trial tests a large number of best-bet technologies or varieties
and is replicated within a site, whereas the baby trials test a smaller subset
of technologies. This facilitates rigorous cross-checking of agronomic
performance with farmers’ assessment. Thus, the approach also allows the
testing of multiple technologies from which individual farmers co-learn and
select the best-bet practices for their own use [28]. Each plot was 20 × 20 m (400
m^2^). Over the course of 3–4 years, the total number of mother
demonstrations was 85 in Kenya, 430 in Tanzania, and 385 in Mozambique. During
this period, the total number of farmers engaged in cereal-pigeonpea
intercropping was estimated at 14,750 in Kenya, 18,000 in Tanzania, and 22,820
in Mozambique.

### Statistical Analysis

2.3

In this analysis we focused on a few of the frequently cited SI indicators, for
which information was available within the scope of the trials. These included
yield, production risk, input use efficiency, and agricultural income, all of
which have been cited as strong SI indicators [29]. As such, the analysis is exclusively based on
productivity and economic indicators.

#### Crop Yield

2.3.1

Among the SI indicator in the literature, crop yield is by far the most
common one [29]. This analysis
focused on cereal grain yield, stover biomass, and wood yield of pigeonpea
across sites in each country. For all analyses, we used a linear mixed
modelling framework because data were unbalanced in terms of number of
treatments and sample sizes. Our focus was on comparison of cereal-pigeonpea
intercropping that received the full recommended fertilizer (Intercrop +
Full) with intercropping without fertilizer application (hereafter called
“Intercrop-Unfertilized”). Wherever available, we also
compared sole maize that received the full recommended fertilizer (Sole
maize + Full) with Intercrop + Full or intercrop that received half of the
recommended fertilizer for the cereal (Intercrop + Half). In all cases, we
used the means and their 95% confidence limit for statistical inference. We
interpreted treatments as significantly different from one another only if
their 95% confidence limits were non-overlapping.

#### Indicators Production Risk

2.3.2

In the context of SI, risk is generally measured as either production risk or
perceived risk [29]. In this
analysis we focused only on production risk. One of the most commonly used
measures of risk is variability, usually indexed by the coefficient of
variation (CV)—a larger CV reflecting more risk [29]. We calculated the CV in
Intercrop + Full, Intercrop-Unfertilized, and Sole maize + Full wherever
applicable. Another common measure of production risk is downside risk,
which can be measured either as the number of years or sites for which
returns or yields are below a target, i.e., deviations from a target. For
this purpose, we estimated the probability of obtaining cereal yield below
the target of 3000 kg ha^−1^ proposed [2]. We estimated this risk measure for maize yield
across sites in Mozambique and Tanzania separately. We also combined all
sites because the maize and pigeonpea varieties used across Mozambique and
Tanzania were more or less the same. Then we plotted the cumulative
probability distribution of yields to aid stochastic dominance analysis, a
non-parametric risk analysis tool often used in decision theory. A
cumulative distribution function of one treatment is said to dominate
another if the first order of its distribution lies entirely to the right of
the cumulative distribution of the other treatment.

#### Indicators of Input Efficiency

2.3.3

Input use efficiency is the other commonly proposed indicator of SI [9]. Among the many metrics of input
efficiency, partial factor productivity (PFP) is the most commonly used SI
indicator [8]. Therefore, we
calculated the PFP of N (PFPN) and P (PFPP) fertilizers applied to the
cereal crop. PFP is measured in kg grain per kg of applied N or P. We also
assessed the agronomic use efficiency (AE) of N and P (i.e., AEN and AEP),
measured in kg grain increase per kg of applied N or P. The AE is an
integrated index of nutrient recovery efficiency and closely reflects the
impact of the applied N or P fertilizer on grain yield. We focused the
comparison of input use efficiency on those treatments where cereals
received N and P inputs, i.e., Intercrop + Full and Intercrop + Half with
Sole maize + Full.

#### Indicators of Agricultural Income

2.3.4

Several metrics of agricultural income are used in the SI literature, but the
most common one is the benefit cost ratio (BCR) [29]. We calculated BCR using data on input prices
(costs of fertilizer, labor, and transport) and outputs (farm-gate of maize
and pigeonpea) acquired through market surveys in each country. We
calculated BCR as the ratio of total returns to total costs for each
treatment at each mother trial. As in the analysis of yields, we used a
linear mixed modelling framework to estimate the mean BCR and its 95%
confidence limits for each treatment. For the adoption of technologies, the
break-even point (BCR = 1) is often not sufficiently attractive, but BCR
significantly larger than 1 is generally considered to be profitable.
Therefore, a treatment was deemed profitable when BCR was significantly
greater than 1, as judged by the width of the 95% confidence limit.

Acceptability of fertilizer by farmers is best judged by the marginal rates
of return (MRR), an approach to maximize profit [30]. Therefore, MRR was computed as the ratio of
the marginal benefit (i.e., the change in net benefits) divided by the
marginal cost (i.e., the change in costs) relative to the intercrop without
fertilizer. As a rule of thumb an MRR less than 50% is considered low and
unacceptable to farmers; a higher cut-off value (e.g., MRR > 100%)
has been recommended if the technology involves significant change from
current farmer practices [30].
Since the application of fertilizer to the intercrop does not represent
significant adjustment to current farmer practice, MRR of 50% was set as an
acceptable minimum in this analysis.

Since pigeonpea can be managed as a perennial crop in cereal cropping systems
[15], investments made in the
first year are expected to provide benefits 2–4 years later.
Therefore, we estimated the net present value (NPV) by discounting and
summing the net benefit for each year. The major assumptions in the NPV
calculation were the time horizon (t) and discount rate. In this analysis, a
5-year time horizon was considered reasonable. For discount rates, NPV
analyses typically use loan interest rates which are set by national banks.
During the project implementation period, interest rates were 14–17%
in Kenya, 15–17% in Tanzania, and 14–19% in Mozambique [31]. Therefore, we used the
country-specific discount rate based on the loan rate as a proxy for the
years 2013 and 2014, which were 17% for Kenya, 16% for Tanzania, and 15% for
Mozambique. If the NPV is positive, the benefits outweigh the costs, and the
investment will generate a profit over time. We also conducted sensitivity
analysis to detect the influence of volatility in output prices on financial
returns. We computed NPV, assuming 10 and 20% fall or rise in maize and
pigeonpea prices while holding cost of inputs constant. We held the cost of
inputs constant and varied the crop prices because crop prices usually
fluctuate widely in a year while input prices often remain constant. We then
compared the outcome with a “no change” scenario in prices. In
order to reduce clutter, we only focused the sensitivity analysis on
intercropping + full fertilizer, which was found to be the best
practice.

## Results

3.

### Cereal Yields, Variability and Risk

3.1

Fertilizer use in intercropping (Intercrop + Full) reduced variability in cereal
yields (CV) by 40–56% and increased yields by 55–294% over
unfertilized intercrops (Intercrop-Unfertilized) across sites in the three
countries (Table 2). Across the sites
in Mozambique, maize grain yield was highest (3267 kg ha^−1^) in
fully fertilized sole maize (Sole maize + Full) but its variability was also
highest (CV = 108%). On the other hand, in the Intercrop + Full, the variability
was lowest (CV = 47.6%), although the mean yield (3165 kg
ha^−1^) was not significantly different from Sole maize + Full
(Figure 1). Yields recorded in
Intercrop-Unfertilized (829 kg ha^−1^) were significantly lower
than in Intercrop + Full, besides being more variable (CV = 107.9%) (Table 2). Across the sites in Tanzania,
maize yield was highest (3130 kg ha^−1^) but its variability was
lowest (CV = 22.2%) in Intercrop + Full. Yields were significantly higher than
in Intercrop-Unfertilized (1625 kg ha^−1^) (Figure 1), which had the highest variability (CV =
45.6%). The probability distribution of yields revealed first order stochastic
dominance of Intercrop + Full over Intercrop-Unfertilized (Figure 2). Across the sites in Mozambique and Tanzania
the probability of exceeding 3000 kg ha^−1^ was significantly
higher in Intercrop + Full than in Intercrop-Unfertilized (Figure 2). When data were combined, the probability was
higher (0.56) in Intercrop + Full compared to 0.04 in
Intercrop-Unfertilized.

**Table 2 t0002:** Cereal and pigeonpea grain yields (in kg ha^—1^), their
coe_cients of variation (CV in %), and percentage change in CV from the
intercrop without fertilizer (Intercrop-Unfertilized).

Crop	Country	Treatment	Mean Yield	CV (%)	% Change in CV
Maize	Mozambique	Intercrop-Unfertilized	829	107.9	
		Intercrop + Full	3165 (282.1) ^[Table-fn tf2-1]^	47.6	−55.9
		Intercrop + Half	2623 (216.6)	57.6	−46.7
		Sole maize + Full	3267 (294.3)	55.8	−48.3
	Tanzania	Intercrop-Unfertilized	1625	45.6	
		Intercrop + Full	3130 (92.6)	22.2	−51.4
		Sole maize + Full	3021 (85.9)	24.2	−47.0
Phosphorus	Mozambique	Intercrop-Unfertilized	1230	59.6	
		Intercrop + Full	2102 (71.0)	35.6	−40.2
		Intercrop + Half	1917 (55.9)	32.7	−45.1
		Intercrop-Unfertilized	252	93.7	
		Intercrop + Full	1384 (449.5)	77.7	−16.6
		Intercrop + Half	1307 (419.1)	88.0	−5.5
		Sole Pigeonpea + Full	3187 (1164.7)	43.4	−53.4
	Tanzania	Intercrop-Unfertilized	346	71.4	
		Intercrop + Full	994 (187.2)	62.1	−13.0
	Kenya	Intercrop-Unfertilized	904	60.6	
		Intercrop + Full	1191 (31.7)	29.0	−52.2
		Intercrop + Half	970 (7.3)	38.3	−36.8

*Figures in parentheses are percentage yield increase over the
Intercrop-Unfertilized.

**Figure 1 f0001:**
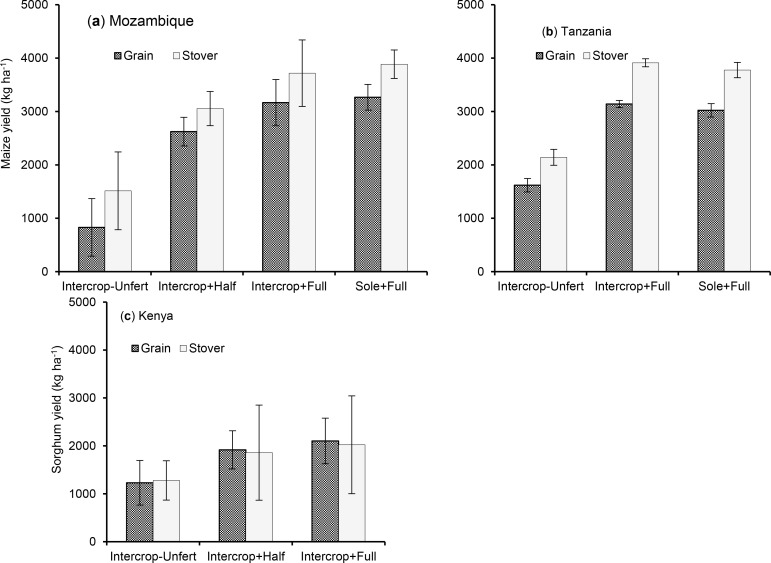
Variation in cereal grain and stover yields with fertilizer application
in cereal-pigeonpea intercropping in Mozambique (**a**),
Tanzania (**b**) and Kenya (**c**). Intercrop-Unfert =
unfertilized cereal-pigeonpea intercrop. Two treatments are deemed
significantly di_erent when their 95% confidence limits (error bars) do
not overlap.

**Figure 2 f0002:**
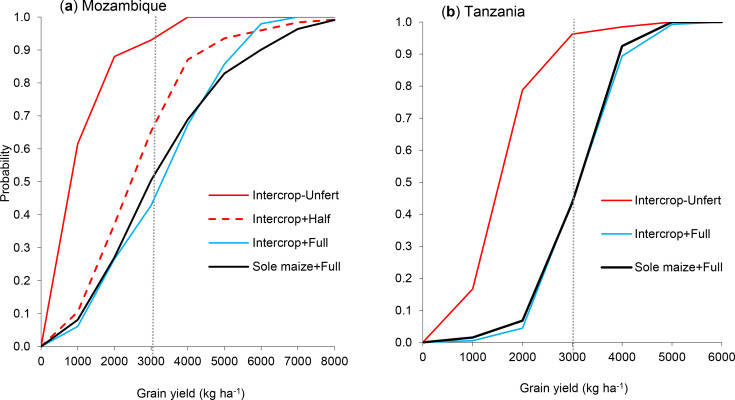
The probability distribution of maize yields in relation to the target
yield of 3000 kg ha-1 in Mozambique (**a**) and Tanzania
(**b**). Yields below the dotted vertical line indicate
production risks.

In Kenya, sorghum yield was significantly higher (2102 kg ha^−1^)
and variability was lower (CV = 35.6%) in Intercrop + Full (Figure 1) compared to Intercrop-Unfertilized (yield =
1230 kg ha^−1^ and CV = 59.6%). Maize and sorghum stover yields
followed the same pattern as grain yield in all countries, with the highest
being in Intercrop + Full (Figure 1).

### Pigeonpea Yields and Variability

3.2

Pigeonpea yields were 31–449% higher in Intercrop + Full over
Intercrop-Unfertilized (Table 2).
Fertilizer use in intercropping reduced variability in yields of pigeonpea by
5–52% over Intercrop-Unfertilized. Across the sites in Mozambique where
sole pigeonpea received N and P fertilizer, grain yield was 3187 kg
ha^−1^ compared to 1384 kg ha^−1^ in
Intercrop + Full and 252 kg ha^−1^ in Intercrop-Unfertilized
(Table 2). In Kenya, Intercrop +
Full gave pigeonpea grain yield of 1191 kg ha^−1^ compared to
904 kg ha^−1^ in Intercrop-Unfertilized and 680 kg
ha^−1^ in sole pigeonpea grown without fertilizer. Intercrop
+ Full fertilizer consistently gave the highest wood yields, ranging from 1729
kg^−1^ ha across sites in Kenya to 7318 kg
ha^−1^ in Mozambique (Figure 3,c). Wood yield measurements were unavailable for sites in
Tanzania (Figure 3).

**Figure 3 f0003:**
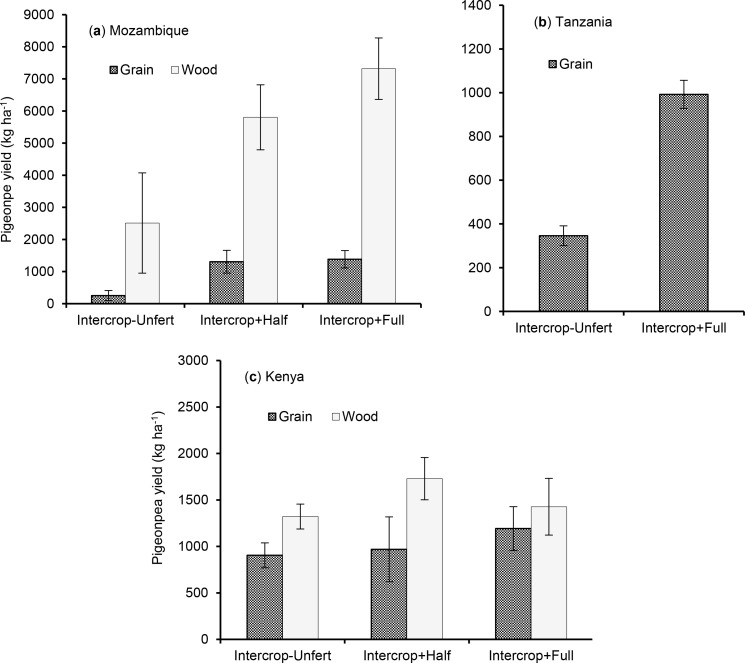
Variation in pigeonpea grain and wood yield with fertilizer rate in
cereal-pigeonpea intercropping in Mozambique (**a**), Tanzania
(**b**) and Kenya (**c**). Intercrop-Unfert =
unfertilized cereal-pigeonpea intercrop. Vertical bars represent the 95%
confidence limits. In Tanzania wood biomass was not determined. Yields
of sole pigeonpea were excluded from figures because these were
available only for some sites

### Input Use Effciency

3.3

The partial factor productivities of N and P were consistently higher (by up to
50%) in Intercrop + Full than Sole maize + Full (Table 3). Across sites in Mozambique partial factor
productivities of N and P were significantly higher in Intercrop + Half than in
Sole maize + Full. AEN was 227% higher in Intercrop + Full than in Sole maize +
Full fertilizer. On the sites in Tanzania, Intercrop + Full and Sole maize +
Full did not significantly differ in both partial factor productivity and
agronomic use effciency of N, but the former consistently had higher values
(Table 3). When data were combined
across sites in Mozambique and Tanzania, AEN and AEP were 37% and 50% higher in
Intercrop + Full than in Sole maize + Full (Table 3). In Kenya, AEN was 57% higher in sorghum-pigeonpea
intercropping that received half of the recommended fertilizer than fully
fertilized sorghum-pigeonpea intercropping. AEP also followed the same trend as
AEN (Table 3).

**Table 3 t0003:** Partial factor productivity (PFP) and agronomic efficiency (AE) of N and
P in sole cereal and in intercropping with pigeonpea with either half or
the full dose of fertilizer recommended for the cereal in Mozambique,
Tanzania, and Kenya.

Nutrient	Country	Treatment	PartialFactorProductivity^[Table-fn tf3-1]^	Agronomic Efficiency ^[Table-fn tf3-2]^
Nitrogen	Mozambique	Intercrop + Half	148.3 (137.7–158.9)	47.1 (38.0–56.2)
		Intercrop + Full	42.9 (26.1–59.7)	16.6 (0.4–56.2)
		Sole maize + Full	34.6 (26.7–42.5)	14.4 (7.6–21.2)
	Tanzania	Intercrop + Full	52.3 (51.2–53.5)	25.5 (24.0–26.9)
		Sole maize + Full	50.4 (48.4–52.3)	25.1 (22.4–27.8)
	Kenya	Intercrop + Half	63.9 (54.2–73.6)	22.9 (13.3–32.5)
		Intercrop + Full	35.0 (25.3–44.8)	14.5 (5.0–24.1)
Phosphorus	Mozambique	Intercrop + Half	145.7 (130.9–160.4)	61.4 (46.7–76.1)
		Intercrop + Full	123.9 (100.4–147.4)	48.7 (22.5–75.0)
		Sole maize + Full	84.8 (73.7–95.8)	35.7 (24.7–46.7)
	Tanzania	Intercrop + Full	157.0 (153.6–160.4)	76.4 (72.1–80.7)
		Sole maize + Full	151.1 (145.2–157.0)	75.3 (67.3–83.3)
	Kenya	Intercrop + Half	191.7 (162.5–220.9)	68.7 (39.9–97.4)
		Intercrop + Full	105.1 (75.9–134.3)	43.6 (14.9–72.3)

*Partial factor productivity is given in kg grain
kg^−1^ of applied N or P.

**Agronomic efficiency is given in kg grain increase
kg^−1^ of applied N or P. Figures in parentheses
represent 95% confidence limits

### Profitability of Fertilizer Use

3.4

The benefit–cost ratios (BCR) were significantly larger than unity in
Intercrop + Full compared to Intercrop-Unfertilized (Table 4). Across sites in Mozambique Intercrop + Half was
more profitable (BCR = 1.4 and MRR = 145) than Intercrop + Full (BCR = 1.1 and
MRR = 38.7). With a BCR of 0.7, Sole maize + Full was as unprofitable as
Intercrop-Unfertilized (Table 4). With
Intercrop + Full, MRR was 178% in Tanzania, meaning that farmers in the study
sites can expect to obtain $2.78 for every $1 investment in purchasing and
applying fertilizer in the intercropping. In Kenya, Intercrop + Half was
slightly more profitable (MRR = 116) than Intercrop + Full (MRR = 82).

**Table 4 t0004:** Benefit–cost ratios (BCR), net present values (NPV), and marginal
rates of return (MRR) in intercropping, with and without fertilizer.

Country	Treatment	BCR	MRR	NPV-5 Year
Mozambique	Intercrop-Unfertilized	0.7 (0.3–1.2) *	NA	−301.1 (−796.3–194.1)
	Intercrop + Full	1.1 (1.0–1.3) ^[Table-fn tf4-1]^	38.7 ^§^	377.3 (^−^6.7–761.2)
	Intercrop + Half	1.4 (1.2–1.5)	145.2	733.5 (338.2–1128.8)
	Sole maize + Full	0.7 (0.6–0.8) ^[Table-fn tf4-1]^	33.7 ^§^	^−^937.7 (^−^1092.7–^−^782.6)
Tanzania	Intercrop-Unfertilized	1.1 (1.0–1.1) ^[Table-fn tf4-1]^	NA	67.2 (^−^55.5–189.9)
	Intercrop + Full	1.5 (1.4–1.6)	177.7	1018.9 (906.9–1130.9)
	Sole maize + Full	1.4 (1.3–1.4)	290.2	574.4 (488.8–659.9)
Kenya	Intercrop-Unfertilized	1.3 (0.9–1.6) ^[Table-fn tf4-1]^	NA	593.3 (^−^156.1–1342.6)
	Intercrop + Full	1.4 (1.2–1.7)	82.1	1192.3 (491.8–1892.9)
	Intercrop + Half	1.4 (1.1–1.7)	116.5	1019.7 (350.6–1688.8)

The figures in parenthesis are 95% confidence limits.

*When the 95% confidence limit encompasses 1, BCR is not significantly
different from 1, and thus the intervention is unprofitable. NA =
not applicable. ^§^ MRR is below the threshold of
50%.

In terms of net present values (NPV), Intercrop + Full was more profitable than
Intercrop-Unfertilized across all countries (Table 4). The NPV was more than 2 times higher in Intercrop + Full
compared with Intercrop-Unfertilized in Tanzania and Mozambique. In Kenya NPV
was 101% higher in Intercrop + Full compared with Intercrop-Unfertilized (Table 4). When all sites were
considered, there was 86% probability of producing maize and pigeonpea
profitably (NPV > 0) in Intercrop + Full. On the other hand, in
Intercrop–Fertilizer there was only 47% probability of producing maize
and pigeonpea profitably.

According to the sensitivity analysis, an increase in maize and pigeonpea prices
by 10–20% will not significantly increase the profitability (i.e., NPV)
of the intercrop in Mozambique and Kenya (Figure 3). A fall in prices by 10–20% will make intercropping
unprofitable in Mozambique. On the other hand only a 20% fall in prices will
make intercropping unprofitable in Kenya. In Tanzania the 95% confidence limits
do not overlap, indicating that intercropping will continue to be profitable
even with a 20% fall in crop prices (Figure 4). Increases in maize and pigeonpea prices by 10% will also
significantly increase NPV.

**Figure 4 f0004:**
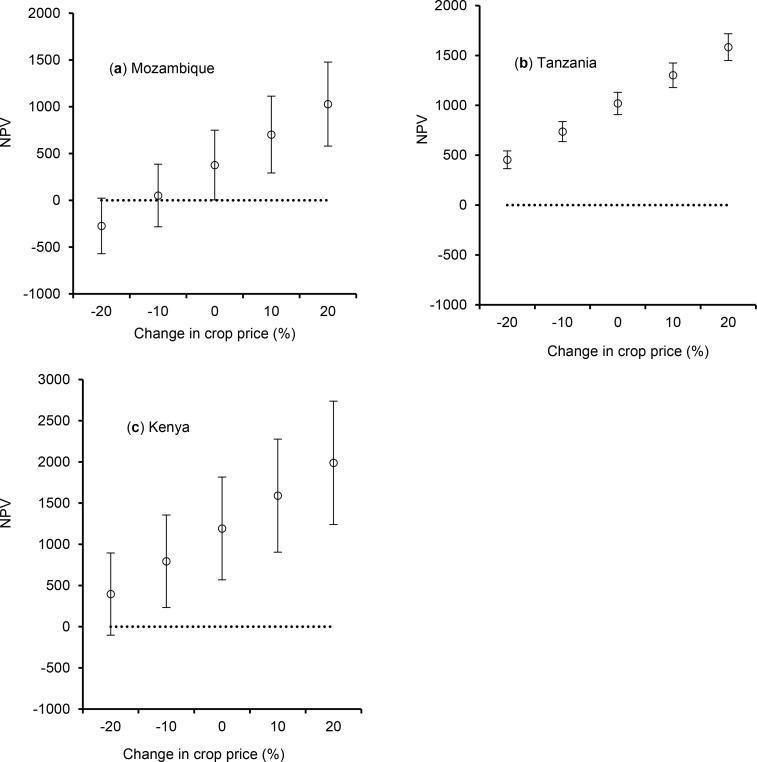
E_ect of price volatility on net present value (NPV) of cereal-pigeonpea
intercropping with the full fertilizer rate in Mozambique
(**a**), Tanzania (**b**) and Kenya
(**c**). When the 95% confidence limit encompasses the
dotted horizontal line, intercropping becomes unprofitable.

## Discussion

4.

All the SI indicators assessed indicated that intercropping combined with the
application of small amounts of fertilizer to the cereals is superior to
intercropping without fertilizer. Indeed, in the absence of fertilizer inputs,
intercropping maize with pigeonpea gives only modest yields. These findings are
consistent with findings from earlier studies in Zimbabwe [26,32] and
Tanzania [20]. These earlier studies have
also reported depression in yields of maize in intercropping relative to monoculture
maize. This has been attributed to competition between maize and pigeonpea for soil
nutrients and water [20]. Our results
also suggest that pigeonpea yields are suppressed in intercropping when fertilizer
is not applied to the maize crop. This is consistent with reports from earlier
studies [20] that showed strong
competition from the maize crop, thus suppressing grain yields of the pigeonpea by
as much as 33%. The increase in grain of pigeonpea is probably because the
fertilizer applied to maize may have benefitted the pigeonpea, since P fertilizer is
crucial for growth and N fixation in legumes. In Mozambique, where N and P were
added to sole pigeonpea, grain yields increased by 3–4 times over the
unfertilized intercrop. Similarly, in Tanzania the addition of N and P fertilizers
has been shown to increase pigeonpea biomass yield by 239% relative to the
unfertilized intercropping treatment [33].

Higher yields with fertilizer use in intercrops implies that farmers could produce
more without putting extra land under cultivation. The increase in yields is
expected to diversify food and income sources for household. With farmgate prices
often 3–4 times more than maize, market opportunities for pigeonpea are
greater than those for maize in the study areas. As such, sales of pigeonpea are
likely to contribute to farmers’ ability to purchase the small quantities of
fertilizers required seasonally. The additional benefit of intercropping is the wood
biomass from pigeonpea harvest, which can be used as a domestic energy source [21]. When pigeonpea production is
integrated with energy-saving stoves, it can also reduce the frequency of buying and
collecting fuel-wood [21]. This could
free some time for women in rural areas who traditionally do the task of firewood
collection for the family. The increase in crop residues (maize or sorghum stover
and pigeonpea wood) due to addition of fertilizer can also add significant benefits
to the farm economy. Crop residues are valuable sources of animal feed and soil
organic matter. For example, stover is increasingly becoming an important source of
soil cover in conservation agriculture (CA). In the absence of fertilizer, average
stover yields were well below the threshold of 3.0 t ha^−1^ required
for good soil cover. Stover yields less than this will not provide the 30% soil
cover required to implement CA successfully [34]. There is also the pigeonpea litterfall,

which although not quantified in this study, could be significant in its contribution
to soil fertility improvement. This could, in the short and medium term, break the
cycle of land degradation and its associated social deprivation issues [35].

Animportantbenefitofapplyingsmallamountsoffertilizersinintercropsisthemoreefficientuseof
the applied N and P fertilizers compared to sole maize. This is probably because
intercropping increases availability of applied nutrients and improves nutrient use
efficiency by associated cereals [20,36]. Fertilizer use is not
only essential to increase crop yields but also replenish the nutrients removed by
crop harvest, and thus sustaining productivity due to a more efficient use of
fertilizers. In many parts of East Africa, the soils are P deficient [2] and its omission has been shown to
depress the attainable yields of maize by 1000–1700 kg ha^−1^
year^−1^ [37].
Without fertilizer application and other good agronomic practices, intercropping
pigeonpea with high yielding varieties of maize is likely to be unsustainable.

Our analyses also indicate that the strategic application of small amounts of
fertilizers (e.g., Intercrop + Half) is profitable in the intercropping systems in
the majority of cases, except where 10–20% fall in maize and pigeonpea prices
makes intercropping unprofitable with application of the fertilizer recommended for
maize. This was the case in Mozambique, where higher input prices (compared to
Tanzania and Kenya) reduced profitability. The positive NPV values in all countries
indicate that overall the benefits of fertilizer use outweigh the costs and the
investment in fertilizer in cereal-pigeonpea intercropping is profitable. As such,
intercropping pigeonpea with cereals can be a good source of cash for resource-poor
and vulnerable households. Good markets exist for pigeonpea products from Africa,
and the demand for processed pigeonpea products outstrips supply by over 30% [16]. For example, the annual pigeonpea
grain exports to India in the last decade recorded a high of about 2500 tons against
an estimated demand of 15,000–20,000 tons [23]. Due to the high demand, farmers in the study sites
were receiving farmgate prices for their pigeonpea crops that were 3–4 times
that of maize. Given this market potential, application of the recommended dose of
fertilizer in intercropping may continue to be profitable even if maize and
pigeonpea prices fall by as much as 20%.

Another important finding from this analysis is the lower yield variability where the
intercrop was amended with fertilizer. Reduced variability implies reduced risk of
income loss, while higher benefit–cost ratio results in higher net income.
The economic analysis suggests the benefit–cost ratios vary with sites, with
some being profitable at half the recommended nitrogen and phosphorus application
ratesof the nationalsystems, which are generally 60and 20 kgha^−1^,
respectively. This should help farmers purchase the small amounts of fertilizers
they need to improve the fertility of their soils and raise its productivity.

Overall, these results indicate that farmers are better off planting pigeonpea in
their maize fields, wherever it is suitable, than simply growing sole maize with the
recommended fertilizer rate. In contrast with the emphasis on cereal monocultures
generally by some development agencies, our results, and those of the growing body
of literature [8,17,38–40], suggest that intercropping can play a central
role in the efforts to achieve climate-smart agriculture. For example, in Malawi and
Mozambique, maize-pigeonpea intercropping without fertilizer application has been
demonstrated to reduce the risk of crop failure and improve profitability [8,17,38,39]. Using historical rainfall records and simulated yield
in Malawi, Snapp et al. [40] showed that
intercropping pigeonpea with maize can meet the household food needs (calories and
proteins) in 73–100% of the years across variable rainfall patterns.
According to Kamanga et al. [38], maize
intercropped with pigeonpea was less risky for resource-poor farmers compared to
fully fertilized maize in central Malawi. Pigeonpea is well adapted to semi-arid
conditions, and tolerant to many environmental stresses [16]. These are important traits for enhancing the
resiliency of smallholder farmers to the growing challenge of climate change and
climate variability. However, there are challenges associated with
*Fusarium* wilt, which was noticeable among pigeonpea stands of
even supposedly tolerant varieties.

Intercropping of cereal with pigeonpea can benefit not only the resource-endowed
farmers but also very poor households. For example, in central Malawi, intercropping
maize with pigeonpea had consistently positive returns across all farmer resource
groups [38]. Similarly, the most
vulnerable households in southern Malawi are better off intercropping pigeonpea with
maize than growing sole maize with the recommended fertilizer [39]. This can lead to higher productivity, as is evident
from our study as well, which could in turn reduce the unit cost of production. This
will not only increase productivity and profitability but it will also accrue other
benefits as it can abate the effects of land degradation, which poses serious
social, economic, and environmental problems in SSA [7]. Recent analysis of SI indicators in Malawi [41] also indicate that social and human
capacity building were superior for pigeonpea maize intercropping rather than
fully-fertilized maize monoculture, notably in terms of dietary diversity, food
security, and farmer preferences.

The challenge has been lack of knowledge and the poor access to fertilizer and seeds
of improved varieties by African farmers. Access to improved varieties of legumes is
limited because production of legume seed is apparently not an attractive venture
for the existing seed companies in the study locations. Policies on input subsidies
also rarely provide for improved legume seed. If input subsidy programs could
broaden their scope to include leguminous crops like pigeonpea, they would be better
able to support the poorest households. There is also an urgent need to invest in
training farmers on good agronomic practices that increase the use efficiency of
inorganic fertilizers in maize-pigeonpea intercropping systems.

## Conclusions

5.

The main conclusion from the present analyses is that the strategic application of
small amounts of fertilizers (15–30 kg N ha^−1^ and
10–20 kg P ha^−1^) along with good agronomic practices is
necessary for cereal-pigeonpea intercropping to be sustainable and financially
profitable in the semiarid regions of ESA. Application of fertilizer would continue
to be profitable even if cereal and pigeonpea prices fall by as much as 20%.
However, without fertilizer use, intercropping alone is likely to suppress pigeonpea
yields, while also increasing nutrient mining when high yielding pigeonpea varieties
are used. Therefore, it is important to encourage farmers to use the appropriate
rates, especially of N and P fertilizers or manure, to the cereal-pigeonpea
intercropping system. Application of small amounts of fertilizers, about half the
levels recommended by the national research organization (for maize yield targets of
3 t ha^−1^ or more), may also help stabilize yields across seasons
and minimize risks of crop loss to the farmers in the event of sub-optimal cropping
seasons. However, there is a need to establish the levels more precisely through
elaborate trials that use fertilizer response curves. Given the generally high cost
of fertilizers in SSA, subsidies that are being promoted in several countries could
be designed to target leguminous crops that have the potential to sustainably
improve food security in the drier agro-ecologies and build more climate-smart and
resilient production systems, such as pigeonpea. An important caution to note is
that when pigeonpea intercropping is scaled-up, diseases (e.g.,
*Fusarium* wilt) and pests may threaten its production and
productivity. Therefore, there is a need for continuous breeding for pest resistance
and high-yielding pigeonpea varieties for areas that are currently under the crop
and others that could potentially be brought under it. Scaling up the production of
pigeonpea intercropping could contribute significantly to household firewood and
energy supply, thus reducing pressure on forests and woodlands in the drylands.

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
