# Peer review of "Sustainable Intensification with Cereal-Legume Intercropping in Eastern and Southern Africa"

_sustainability, 2019, doi:10.3390/su11102891_

Reviewer 1 Report

Sustainable intensification (SI) with intercropping is a very interesting topic. The efficiency in resource use is a crescent need for sustainable development of agriculture.

I understand that this MS deals with fertilization use in intercropping systems in several farms in three countries.

In my opinion, this MS fails in quantify the amount of fertilization that could increase SI indicators in a sustainable way. 

The use of fertilization and increase in yield is already well study.

Suggestions to improve the MS are written in the text.

Author Response

Sustainable intensification (SI) with intercropping is a very interesting topic. The efficiency in resource use is a crescent need for sustainable development of agriculture. I understand that this MS deals with fertilization use in intercropping systems in several farms in three countries. In my opinion, this MS fails in quantify the amount of fertilization that could increase SI indicators in a sustainable way. The use of fertilization and increase in yield is already well study. Suggestions to improve the MS are written in the text.

Response: We could not quantify the amount of fertilization that could increase SI indicators in a sustainable way because of the lack of data. We believe, that requires much more data and developing nutrient response functions for each site, which is beyond the scope of this study. However, we have recommended 15-30 kg N ha-1 and 10-15 kg P ha-1 as a reasonable range. We also recommended this to be refined through future area of work on relevant sites. We have thoroughly revised taking into account all the suggestions annotated on the PDF version of the manuscript.

Line 46: We have reversed the references

Line 56: We have provided the species name for pigeonpea

Line 59: We have provided the species names for all plants mentioned

Line 66: We have revised the references as suggested

Line 74: We have revised the references as suggested

Line 75: We have corrected this error throughout the manuscript as suggested

Line 96: We have written SSA in full as sub-Saharan Africa.

Line 119: Yes. In this analysis cereal-pigeonpea intercropping without fertilizer was used as the control, against which all other treatments were compared.

Line 140: We have uniformly applied kg ha-1 throughout the manuscript.

Line 226: We have corrected this reference

Line 229: We have now deleted the dot following MRR

Line 235: We have corrected the font..

Line 401: By small amounts we mean about 15-30 kg N ha-1 and 10-15 kg P ha-1. We have now indicated that in the manuscript

Line 417: We have corrected this to pigeonpea.

Reviewer 2 Report

The paper is interesting and in line with other contributions in the SI literature.

It is on the whole balanced, but it seems that it wants to compare the different territorial contexts in Africa. 

Only in the spirit of improvement would I make the following revisions:

- I would like to better specify the cultural differences and the socio-economic situations in the geographical contexts under study and the possible elements of homogeneity?;

- what the system effect is?;

- what is the agroecological approach followed in the different countries, also to assess the conditions of residual fertility of the land;

- some tests were conducted in 2012-2014 (line 121) and others in 2010-2012 (line 136): why all this time to obtain results to be published? how do climate aspects influence the tests?

- the calculation of the net present value assumes the collection of cash flows (income and expenditure): has a single accounting system been set up for all realities?

Author Response

Comments and Suggestions for Authors

The paper is interesting and in line with other contributions in the SI literature. It is on the whole balanced, but it seems that it wants to compare the different territorial contexts in Africa. Only in the spirit of improvement would I make the following revisions:

- I would like to better specify the cultural differences and the socio-economic situations in the geographical contexts under study and the possible elements of homogeneity?;

Response: We have now addressed this at the beginning of Section 2.1 in the revised manuscript

- what the system effect is?;

Response: We have now addressed this at the beginning of Section 2.1 in the revised manuscript

- what is the agroecological approach followed in the different countries, also to assess the conditions of residual fertility of the land;

Response: We have now addressed this in terms of the farming system in the revised manuscript

- some tests were conducted in 2012-2014 (line 121) and others in 2010-2012 (line 136): why all this time to obtain results to be published? how do climate aspects influence the tests?

Response: This longer time period was necessary to generate adequate data for the kind of analysis we conducted. In order to avoid confounding of results by climate differences we conducted the analysis for each country separately, but site differences within country were accounted for through the use of the mixed linear modelling approach.

- the calculation of the net present value assumes the collection of cash flows (income and expenditure): has a single accounting system been set up for all realities?

Response: All calculations were made using standard methods using a single accounting system.

Reviewer 3 Report

GENERAL COMMENTS

Based on the sustainable intensification approach, the article presents results of the economic and productivity benefits of the use of fertilizers in cereal-pigeonpea intercropping.

Nevertheless, sustainability is a very broad concept that in addition to economic and productivity indicators and metrics includes other important aspects associated with environmental, social, human wellbeing and gender equity indicators. Therefore, I suggest that throughout manuscript it is emphasized that the obtained results are based exclusively on economic and productivity indicators.

In my personal opinion, using the name Intercrop-Fertilizer for treatment intercrop without fertilizer throughout the manuscript is confusing. I suggest change the name of this treatment by Intercrop-Unfertilized o Intercrop-No Fertilizer.

Introduction

In this section, the authors would include more information on evaluation methodologies of the SI approach.

Materials and Methods

In this section, the authors should include information on plant densities per hectare, planting methods, and in general on the agronomic practices carried out in the different trial sites.

Also, please clarify which sources of N and P were used in Mozambique and Kenya

Discussion

I am not satisfied with the overall discussion of the manuscript. In this section, I consider that the article could be benefit if, based on recent literature, the authors discuss the methodological limitations or the results of the study (for example, the number and type of SI indicators used). The authors could also compare the results of the study with the results obtained in other IS studies using other indicators and metrics.

Tables and Figures

In Table 2, the treatment Sole-pigeonpea + Fertilizer in Mozambique also is confusing. Does it refer to Sole-pigeonpea + Full or Sole-pigeonpea + Half treatments? Please clarify.

In Tables 2; 4; Figures 1a, 1b, 1c; Figures 2a, 2b, 2c; Figures 3a, 3b, 3c; using the name Intercrop-Fertilizer for treatment intercrop without fertilizer is confusing. I suggest change the name of this treatment by Intercrop-Unfertilized or Intercrop-No Fertilizer.

Table 3. In this Table, no results are presented for sorghum crop, so this word should be eliminated in Table title. On the other hand, what are the units of the partial factor productivity and agronomic efficiency? Also clarify what is the meaning of the values in parentheses in the columns Partial factor productivity and Agronomic efficiency.

Figure 1. In Figure 1b, I suggest reverse the order of the columns Sole + Full and Intercrop + Full.

References

The format and writing of several references should be revised.  Recent journal issues should be studied in this regard.

SPECIFIC COMMENTS:

Line 23. Replace this “of fertilizer…” by “of inorganic fertilizer …”

Line 30. Replace this “of fertilizers…” by “of inorganic fertilizers …”

Line 31. Replace this “for the sustainability of cereal-pigeonpea…” by “for the productivity and economic sustainability of cereal-pigeonpea…”

Line 46. Replace this “[1,1,8,9].” By “[1,8,9].”

Line 96. What is the meaning of SSA?

Lines 120-133. In this paragraph, please clarify which were the N and P fertilizers used in Mozambique.

Lines 145-154. In this paragraph, please clarify which were the N and P fertilizers used in Kenya.

Line 172. Replace this “yield production risk, input use efficiency…” by “yield, production risk, input use efficiency…”

Line 220. Replace “95% CL for…” by “95% confidence limits (CL) for…” . Throughout the manuscript, review and standardize the use of terms confidence intervals and confidence limits, as well as their acronyms.

Line 382. Replace this “with Fusarium wilt…” by “with Fusarium wilt…” (Use Italic letters).

Line 414. Replace this “e.g. fusarium wilt…” by “e.g. Fusarium wilt…” (Use Italic letters).

Lines 441-442. The format and writing of this reference should be revised; correct the use of semicolon in this reference.

Line 464. The year should be in bold letters; indicate the initial and final pages of the article.

Lines 495-496. The format and writing of this reference should be revised; the year must be in bold letters, after the name of the Journal.

Lines 497-499. The format and writing of this reference should be revised; the year must be in bold letters, after the name of the Journal.

Lines 509-510. Complete this reference; indicate volume, initial and final pages of the article.

Lines 536-537. The format and writing of this reference should be revised; the year must be in bold letters, after the name of the Journal.

Lines 540-545. Table 1. In this Table, I suggest adding an additional column with the condition or fertility level of the annotated soils; also, I suggest ordering the names of the countries and information of the sites as presented in Tables 2 and 3 (Mozambique, Tanzania and Kenya) (as also are presented throughout the manuscript).

Line 557. Replace this “(maize or sorhum and…” by “maize and…”

Lines 556-559. Table 3. Partial factor productivity and Agronomic efficiency columns. Indicate the units of both indicators.

Lines 567-570. Figure 1b. In this Figure, I suggest inverting the order of columns Sole+Full and Intercrop+Full

Author Response

GENERAL COMMENTS

Based on the sustainable intensification approach, the article presents results of the economic and productivity benefits of the use of fertilizers in cereal-pigeonpea intercropping. Nevertheless, sustainability is a very broad concept that in addition to economic and productivity indicators and metrics includes other important aspects associated with environmental, social, human wellbeing and gender equity indicators. Therefore, I suggest that throughout manuscript it is emphasized that the obtained results are based exclusively on economic and productivity indicators.

Response: In section 2.3 of the revised manuscript we have indicated that the analysis is exclusively based on economic and productivity indicators.

In my personal opinion, using the name Intercrop-Fertilizer for treatment intercrop without fertilizer throughout the manuscript is confusing. I suggest change the name of this treatment by Intercrop-Unfertilized o Intercrop-No Fertilizer.

Response: Throughout the manuscript we have replaced Intercrop-Fertilizer with Intercrop-Unfertilized.

Introduction

In this section, the authors would include more information on evaluation methodologies of the SI approach.

Response: We have now provided information on evaluation methodologies of the SI approach just before the objectives of the study.

Materials and Methods

In this section, the authors should include information on plant densities per hectare, planting methods, and in general on the agronomic practices carried out in the different trial sites.

Response: We have now provided information on the recommended agronomic practices at the end of section 2.1.

Also, please clarify which sources of N and P were used in Mozambique and Kenya

Response: We have now clarified that. The N fertilizer was applied in the form of calcium ammonium nitrate (CAN), while the phosphorus fertilizer was applied in the form of DAP

Discussion

I am not satisfied with the overall discussion of the manuscript. In this section, I consider that the article could be benefit if, based on recent literature, the authors discuss the methodological limitations or the results of the study (for example, the number and type of SI indicators used). The authors could also compare the results of the study with the results obtained in other IS studies using other indicators and metrics.

Response: We have now expanded the discussion making reference to results of other studies. However, we found only one more study that applied SI indicators, namely Snapp et al. (2018). We have now included that reference. 

Tables and Figures

In Table 2, the treatment Sole-pigeonpea + Fertilizer in Mozambique also is confusing. Does it refer to Sole-pigeonpea + Full or Sole-pigeonpea + Half treatments? Please clarify.

Response: This refers to sole pigeonpea + Full. We have now corrected this error

In Tables 2; 4; Figures 1a, 1b, 1c; Figures 2a, 2b, 2c; Figures 3a, 3b, 3c; using the name Intercrop-Fertilizer for treatment intercrop without fertilizer is confusing. I suggest change the name of this treatment by Intercrop-Unfertilized or Intercrop-No Fertilizer.

Response: We have now replaced Intercrop-Fertilizer with Intercrop-Unfertilized throughout the manuscript. 

Table 3. In this Table, no results are presented for sorghum crop, so this word should be eliminated in Table title. On the other hand, what are the units of the partial factor productivity and agronomic efficiency? Also clarify what is the meaning of the values in parentheses in the columns Partial factor productivity and Agronomic efficiency.

Response: The results presented under Kenya are for sorghum. We have now indicated at the bottom of the table the units of partial factor productivity and agronomic efficiency. We have also indicated that fFigures in parentheses represent 95% confidence limits.

Figure 1. In Figure 1b, I suggest reverse the order of the columns Sole + Full and Intercrop + Full.

Response: We have now reversed the order Sole + Full and Intercrop + Full.

References

The format and writing of several references should be revised.  Recent journal issues should be studied in this regard.

Response: We have carefully revised the reference following the style in a recent journal issue. 

SPECIFIC COMMENTS:

Line 23. Replace this “of fertilizer…” by “of inorganic fertilizer …”

Response: We have done this in the revised manuscript.

Line 30. Replace this “of fertilizers…” by “of inorganic fertilizers …”

Response: We have done this in the revised manuscript.

Line 31. Replace this “for the sustainability of cereal-pigeonpea…” by “for the productivity and economic sustainability of cereal-pigeonpea…”

Response: We have now replaced this in the revised manuscript.

Line 46. Replace this “[1,1,8,9].” By “[1,8,9].”

Response: We have replaced this in the revised manuscript.

Line 96. What is the meaning of SSA?

Response: We have now written this in full as Sub-Saharan Africa.

Lines 120-133. In this paragraph, please clarify which were the N and P fertilizers used in Mozambique.

Response: We have done this in the revised manuscript.

Lines 145-154. In this paragraph, please clarify which were the N and P fertilizers used in Kenya.

Response: We have now clarified that. The N fertilizer was applied in the form of calcium ammonium nitrate (CAN), while the phosphorus fertilizer was applied in the form of DAP

Line 172. Replace this “yield production risk, input use efficiency…” by “yield, production risk, input use efficiency…”

Response: We have done this in the revised manuscript.

Line 220. Replace “95% CL for…” by “95% confidence limits (CL) for…” . Throughout the manuscript, review and standardize the use of terms confidence intervals and confidence limits, as well as their acronyms.

Response: We have now replaced this phrase.

Line 382. Replace this “with Fusarium wilt…” by “with Fusarium wilt…” (Use Italic letters).

Response: We have now replaced this as suggested 

Line 414. Replace this “e.g. fusarium wilt…” by “e.g. Fusarium wilt…” (Use Italic letters).

Response: We have now replaced this as suggested 

Lines 441-442. The format and writing of this reference should be revised; correct the use of semicolon in this reference.

Response: We have now corrected this reference following the journal style 

Line 464. The year should be in bold letters; indicate the initial and final pages of the article.

Response: We have now corrected this reference following the journal style 

Lines 495-496. The format and writing of this reference should be revised; the year must be in bold letters, after the name of the Journal.

Response: We have now corrected this reference

Lines 497-499. The format and writing of this reference should be revised; the year must be in bold letters, after the name of the Journal.

Response: We have now corrected this reference

Lines 509-510. Complete this reference; indicate volume, initial and final pages of the article.

Response: We have now corrected this reference

Lines 536-537. The format and writing of this reference should be revised; the year must be in bold letters, after the name of the Journal.

Response: We have now corrected this reference

Lines 540-545. Table 1. In this Table, I suggest adding an additional column with the condition or fertility level of the annotated soils; also, I suggest ordering the names of the countries and information of the sites as presented in Tables 2 and 3 (Mozambique, Tanzania and Kenya) (as also are presented throughout the manuscript).

Response: We have now ordered the names of the countries as in Table 2 and 3. But we did not insert information on the condition or fertility level of the soils because the same soil type (e.g. Lixisols) can have different fertility levels depending on the soil management, climate and land use. We believe inserting an assumed average fertility level can give a misleading picture.

Line 557. Replace this “(maize or sorhum and…” by “maize and…”

Response: We have now corrected this in the revised manuscript

Lines 556-559. Table 3. Partial factor productivity and Agronomic efficiency columns. Indicate the units of both indicators.

Response: We have now indicated the units as note at the bottom of the table. We have also written the units in full in the body of the text under section 2.3.3.

Lines 567-570. Figure 1b. In this Figure, I suggest inverting the order of columns Sole+Full and Intercrop+Full

Response: We have now rearranged the figures as suggested.

Round  2

Reviewer 1 Report

After all the review process the MS get improved.

I have little suggestions to give and it is all on the pdf version.

Line 309 - Change "&" for "and"

Line 478 - Correct "writ-up" for "write-up"

In the reference list withdraw or include DOI number of all of the references. Standardize.

Author Response

Line 309 - Change "&" for "and"

Response: We have now corrected this.

Line 478 - Correct "writ-up" for "write-up"

Response: We have now corrected this.

In the reference list withdraw or include DOI number of all of the references. Standardize.

Response: We have now deleted the DOI for the single reference cited.

Reviewer 3 Report

SPECIFIC COMMENTS:

Line 31. Replace this “for the productivity and sustainability of …” by “for the productivity and economic sustainability of …”

Lines 147-159. In this paragraph, it is necessary to include the sources of N: calcium ammonium nitrate (CAN) and P: diammonium phosphate (DA)

Line 463. Replace this “(e.g. fusarium wilt)” by “(e.g. Fusarium wilt)”

Lines 557-558. In this reference, please include the number pages.

Line 587. Replace this “Snappa, S.; Grabowskia, P.; Chikowoa, R.; …” by “Snapp, S.; Grabowski, P.; Chikowo, R.; …”

Author Response

Line 31. Replace this “for the productivity and sustainability of …” by “for the productivity and economic sustainability of …”

Response: We have now corrected this line 

Lines 147-159. In this paragraph, it is necessary to include the sources of N: calcium ammonium nitrate (CAN) and P: diammonium phosphate (DA)

Response: We have now included this in line 158-159.

Line 463. Replace this “(e.g. fusarium wilt)” by “(e.g. Fusarium wilt)”

 Response: We have now replaced this with Fusarium wilt

Lines 557-558. In this reference, please include the number pages.

 Response: We have now provided page numbers

Line 587. Replace this “Snappa, S.; Grabowskia, P.; Chikowoa, R.; …” by “Snapp, S.; Grabowski, P.; Chikowo, R.; …”

Response: We have now corrected these errors